# How Does Digital Transformation Improve Supply Chain Performance: A Manufacturer's Perspective

**Jae Wook Kim [1], Jin Hwa Rhee [2],* and Chul Hung Park [3]**

[1]  Business School, Korea University, 145 Anam-ro, Seongbuk-gu, Seoul 02841, Republic of Korea; jaewook@korea.ac.kr
[2]  College of Business, Daegu University, 201 Daegudae-ro, Gyeongsan-si 38453, Republic of Korea
[3]  Mahlkonig Korea, #1102, 109 Mapo-daero, Mapo-gu, Seoul 04781, Republic of Korea; chul3522@gmail.com
*  Correspondence: jhrhee@daegu.ac.kr; Tel.: +82-53-750-6233

**Abstract:** A prominent research area pertains to the integration of digital technologies in corporate frameworks and their strategic utilization. In particular, as both intercompany dependencies in business activities and environmental uncertainty increase, digital transformation has become an important means of managing transaction relationships not only within but also between companies. The purpose of this study is to explicate the process of how digital transformation technology used among supply chain members can improve corporate performance and to identify the influencing variables for making good use of it. The findings have implications that can help companies invest time and money in digital innovation to achieve effective corporate performance. This research model analyzed data from 222 domestic manufacturing companies through structural equation model analysis. We found that the more developed the corporate culture and the higher the trust with partner companies, the more active the companies are in utilizing digital transformation. In addition, while digital transformation has a direct impact on corporate performance, we also confirmed the mediating effect of information sharing between companies, which can have a greater positive impact on corporate performance as its level increases. A notable result is that digital transformation significantly improves information sharing in low-trust corporate relationships. These results suggest that digital, non-face-to-face technologies can complement and strengthen relationships that have traditionally been formed through interpersonal relationships. This study compensates for the shortcomings of previous studies that verify the fragmentary achievements of digital transformation. It also has theoretical significance in that it hypothesizes and demonstrates the entire process of how digital transformation is activated in what type of environment and leads to corporate performance. In addition, although companies with a strong relationship of trust may find it easy to invest in innovation, there are practical implications that even new companies that do not have a relationship of trust should consider active use of digital transformation when conducting important transactions.

**Keywords:** digital transformation; supply chain; corporate culture; B2B trust; time-based performance

## 1. Introduction

In today's corporate strategy, one of the biggest topics is how companies apply and utilize digital technology in their business sites [1,2]. According to IDG (International Data Group, Inc., Needham, MA, USA)'s 2023 report [3], 89% of companies have adopted or plan to adopt a digital-first strategy, and this strategic trend is becoming just as important for manufacturing companies as well as service companies. This research report explains two main reasons for this change. First, with the development of technology, competition between companies has intensified, and the needs of end consumers have become more diverse and rapidly changing, increasing environmental uncertainty. This means that the competitiveness of an individual company does not come from the company alone but can be acquired and maintained through cooperation among supply chain members [4–6].

Competitiveness obtained in this way allows the company to survive and achieve financial growth, which in turn is an important factor in the sustainability of the company. Corporate sustainability refers to the ability to continuously satisfy the needs of corporate shareholders [7–9]. This corresponds to the economic perspective among the three pillars (environmental, economic, and social) of sustainability mentioned by reference [10], which can be understood as creating new value through cost reduction or differentiation of the company. In other words, in order to maintain corporate sustainability in a business environment with high uncertainty, the use of digital technology is emerging as a means of efficient and effective cooperation and communication between companies [11–13]. The second reason is that with the advancement of IT technology, the technologies that companies can utilize have become easier to use and more diverse. In particular, users (workers) have become accustomed to the digital living environment due to COVID-19 [14], which created a need for people to work from home or make online purchases and, in turn, led to increased use of mobile devices and computers regardless of gender or age.

Prior research has presented various concepts related to digital strategies that can be applied to corporate systems [15–17]. Specifically, 'digitization' refers to digitizing, storing, and utilizing analog data [16], and 'digitalization' refers to incorporating digital technology into an operating system beyond digitizing information [18]. 'Digital transformation' means promoting corporate innovation by embracing digital technology throughout the organization's management and corporate operating system [16]. The literature shows generally positive results regarding the performance of these digital strategies [1,2]. Researchers argue that digitized information is efficiently stored, disseminated, and analyzed and is effective in finding new business opportunities and creating new corporate value.

However, despite the positive research results, the adoption of digital technologies in business does not always guarantee positive outcomes [19], and the level of digital transformation of companies is also bound to vary [20]. Understanding the complexity of the application and implementation process of the strategy is very important before this strategy can lead to improved competitiveness and increased profitability of the company [21]. Based on a survey by reference [22] pointed out that 70% of companies that adopted digital strategies failed to achieve their adoption goals, which meant economic losses amounting to USD 900 billion. The researchers suggested a variety of reasons for this failure, including inappropriate resource allocation, lack of budget, and poor leadership.

Ultimately, solving these problems requires deeply exploring the relationship between digital transformation and corporate performance. However, until recently, related studies have failed to reveal in detail the role and influence of digital transformation in increasing corporate sustainability [23–26]. Studies that have found positive effects of digital transformation only verify fragmentary effects such as information sharing, cooperative attitude, cost reduction, and operational efficiency. There is not much exploration into the broader understanding of the process and the antecedent or moderating effects of various factors of the business environment.

Therefore, the purpose of this study is to focus on the relationship between manufacturing companies and other companies among supply chain members and explore how digital transformation technologies used between them can be effectively introduced and utilized. Specifically, based on previous studies, we explain a company's developmental corporate culture and the concept of trust between companies as antecedent variables for companies to accept digital transformation [27–29]. Next, based on the fact that digital strategy has the basic goal of efficient information sharing and dissemination between firms through the digitization of information [12,13,30], we study information sharing as a parameter to explain the performance of digital transformation. Lastly, we assume that the stronger the trust relationship between companies, the higher the level of information sharing between companies through digital transformation, thereby further improving corporate performance. This study collected and analyzed surveys from 222 Korean manufacturing companies to verify the research hypotheses.

In the next section, we summarize previous studies that will support the research model and propose our research hypotheses. We then introduce the data collection process and the results of hypothesis testing using a structural equation model. Finally, we summarize our findings and discuss implications and limitations of the study.

## 2. Theory and Hypotheses

### 2.1. Digital Transformation

The digital strategies that companies apply to their internal operating systems and B2B (business-to-business) transactions (supply chain) can be distinguished based on their scope and level [16]. At the most basic level, 'digitization' refers to converting analog information into digital bits for utilization [30]. Digital data created in this way has the advantage of being easier to store, transmit, utilize, and analyze compared to analog data [31]. Next, 'digitalization' involves introducing digital technology into the company's overall business processes to establish an efficient operational system [18,32]. For example, companies can significantly reduce the time allocated to paper-based tasks by installing an in-house intranet to utilize electronic documents and emails. 'Digital transformation' implies the application of digital technology to various aspects of organizational management, products, asset management, and operational processes [15,33]. In other words, digital transformation is the introduction of a new business model by implementing digital technology [16] through which companies can develop new core competencies and secure competitive advantage [34].

In particular, companies need to actively consider digital transformation not only for themselves but also for the transaction relationships between companies in the supply chain. In the current environment, business activities are carried out through numerous interactions between companies. So, the confusion or difficulties of one company in the supply chain quickly affect many companies in the supply chain [6]. Therefore, how transaction relationships between partner companies are managed will have a significant impact on corporate performance in today's highly uncertain business environment [5,35]. For example, the digital transformation between manufacturing and distribution companies in the supply chain enables the following expected effects: Through a digital system that enables efficient collaboration with distributors, manufacturers can receive more accurate market information and customer feedback from distributors in a timely manner and reflect it in the development team's new product strategy. In addition, the production team of a manufacturing company can establish an efficient production plan by sharing product shipment quantity and expected demand information with the sales team through a digital IT system. Sharing resource information based on this production plan with the procurement team allows for more accurate ordering, which means that new products that satisfy customers can be quickly developed and brought to market. Ref. [36] uses data from Chinese manufacturing companies to show that the digitalization of the supply chain increases the cost efficiency of companies and increases their information and communication efficiency, thereby flexibly responding to many risks that may appear in an uncertain trading environment and improving corporate performance.

### 2.2. Information Sharing

Fundamentally, corporate digital strategies, including digital transformation, start by converting analog data into digital bits with the goal of efficiently disseminating and sharing digitized information. Specifically, companies adopt digital technologies to manage their supply chains [37], thereby strengthening connections between companies [38], which supports the integration of information between companies [39]. Therefore, information sharing is a crucial variable in the study of the effectiveness of digital transformation.

Information sharing is defined as the extent to which critical and proprietary information is communicated among supply chain members. Information sharing has been of interest in the field of economics [40–43] and has gained significant attention in management strategies, especially in supply chain management (SCM) research from the 2000s

onwards [39,44–49]. Previous studies on information sharing in SCM have variously verified the effectiveness of information exchange. For instance, operational information sharing can leverage economies of scale, and strategic information sharing can engender novel value creation across interconnected parties [50]. Cooperation between SCM members can be improved by sharing production and delivery schedules [51]. A manufacturing company's ability to respond to problems can be improved by sharing information about risks and problems that arise during the resource procurement process with suppliers [52]. Manufacturing companies can overcome the crisis of unsustainable supply chains by sharing customer information with suppliers through information technology [53]. In addition, many studies emphasize that by sharing cost information or market demand information between companies, manufacturing companies can enhance their product innovation capabilities while distribution companies are able to achieve efficient inventory management.

*2.3. Time-Based Performance*

The primary expected effect of digital transformation and information sharing is the enhancement of operational efficiency within companies, enabling swift resolution of tasks or goals [54–57]. The timely execution of tasks is a critical factor for corporate growth and innovation [58,59]. Therefore, we are utilizing 'Time-based performance' as the outcome variable for this study. Ref. [60] defines time-based performance by dividing it into three subconcepts. Time-to-market refers to the time taken for new product development and launch. Time-to-product is defined as the time it takes for procurement and production activities, and responsiveness signifies the time taken for customer service and product support (e.g., after-sales service, installation, training, etc.).

As explained in Sections 2.1 and 2.2 above, prior research on digital transformation between firms has shown that it enhances various transaction outcomes by activating information exchange among companies. Ref. [11] shows that through IT technology, supply chain members (customer, manufacturer, and logistics provider) can share real-time transportation information, thus reducing boundaries between companies and enabling efficient collaboration. Ref. [61] also argues that information sharing with suppliers is positive for the corporate performance of manufacturing companies and the maintenance of business relationships by reducing various transaction risk factors. This means that the digital transformation of two companies working in different locations and in different ways will greatly change the level of information sharing by eliminating time and space constraints by utilizing common digital forms and the internet. Therefore, this study proposes the following:

**H1.** *Information sharing (between firms) positively mediates the relationship between digital transformation and time-based performance.*

Digitizing a significant portion of a company's information for digital transformation between companies can have a direct positive impact on performance even without information sharing among firms. This is because digitized information is useful for being transmitted, spread, and utilized within companies. In particular, time-based performance can be achieved when information obtained from outside the company is effectively utilized within the company. Therefore, it is assumed that the existence of digital transformation between companies can directly affect performance by increasing work efficiency within the company. Thus, we predict the following:

**H2.** *Digital transformation (between firms) positively affects time-based performance.*

*2.4. Corporate Culture*

Although digital integration is important, not all companies embrace it or achieve great results. As digital integration is an innovation that requires considerable cost, time,

and effort from members, companies must explore under what conditions this strategy can be well accepted and produce good results.

Corporate culture encompasses the shared and enduring values and concepts specific to a company that is embraced and enjoyed by members of the organization over the long term [29]. This includes organizational structure, systems, and ethical standards. The application of digital technology throughout the operational processes and the transformation of business processes and systems, which is the focus of this study, represents a significant innovation from a traditional organizational perspective. In the same context, organizational culture has been studied as an important variable affecting the changing corporate environment [27] and innovation [62].

Prior research has utilized various methods to typify and utilize organizational culture in studies. In various studies in the fields of management and economics, there is no unified definition or distinction for corporate culture, and it has been used with various meanings and concepts [63]. Ref. [64] divides organizational culture into a conductive culture that quickly accepts changes in the external environment and is active in internal integration and a dominant culture that maintains the status quo and prioritizes short-term performance. It is argued that a conductive culture is effective in a company's innovation ability (introducing new technologies, developing market-leading products, discovering business opportunities, responding to new customer demands, investing in R&D, etc.). Ref. [65] says that a risk-taking organizational culture is more active and faster than a risk-aversion organizational culture in accepting and adapting to new technologies that involve high uncertainty. Refs. [62,63] show that a creative corporate culture that flexibly accepts employees' thoughts and actions is more advantageous in accepting new technologies or suggesting new processes and logistics systems.

Most of the existing studies that conceptualize organizational culture in various ways refer to the work of reference [66], who categorized corporate culture into developmental culture and hierarchical culture. A developmental culture shows flexible and extroverted characteristics, challenges new business opportunities, emphasizes innovation, and is active in R&D and new product launches. On the other hand, a hierarchical culture emphasizes stability and efficiency, preferring formalized rules and procedures. We apply this concept to propose the following hypothesis.

**H3.** *Developmental corporate culture positively affects digital transformation (between firms).*

*2.5. Business-to-Business (B2B) Trust*

If corporate culture is a characteristic within a company, it is necessary to also pay attention to variables related to the characteristics of relationships between companies. Previous studies on B2B relationship marketing explain the characteristics of B2B transaction relationships through various variables such as satisfaction, trust, commitment, and opportunism [67–69]. Trust, defined as "a willingness to rely on an exchange partner in whom one has confidence" [70], has been most commonly used in previous studies of B2B relationship marketing.

According to agency theory, companies engage in transactions with other firms within the supply chain to carry out managerial activities. In these cases, two companies with differing goals and preferences prioritize their own interests, giving rise to numerous potential conflicts (performance/cost allocation, information asymmetry, opportunistic behavior, etc.) within the relationship [28,71,72]. According to this theory, companies consider information as a tradable commodity that they invest resources and effort to acquire. Therefore, profit-seeking companies are defensive about providing information, leading to imbalances and inequalities in information between companies [28]. In essence, inter-firm information sharing does not naturally occur and is possible only under specific transactional circumstances. In other words, in order to effectively carry out corporate innovations such as digital transformation with other companies, a mutually cooperative attitude between the two companies must be assumed [73]. Ref. [74] explains that a

collaborative attitude between companies can be formed based on trust and commitment. They suggest that in committed relationships, relational investments are accepted, and when there is mutual trust, companies are willing to engage in new innovative challenges and bear the associated risks.

Based on these previous studies, we can assume that B2B trust will be an antecedent variable for digital transformation that will promote information sharing between companies. It is also expected to be a moderating variable that strengthens the positive effect of digital integration on information sharing. Thus, we predict the following:

**H4.** *B2B trust positively affects digital transformation (between firms).*

**H5.** *B2B trust positively affects the relationship between digital transformation (between firms) and information sharing (between firms).*

## 3. Methods and Analysis

### 3.1. Sample and Data Collection

To examine the hypotheses, we acquired data from manufacturing firms in South Korea. For data gathering, we recognize the importance of surveying senior executives, such as the supply chain manager, CEO, and director, who possess comprehensive knowledge of the company's overall operations. Nevertheless, due to the low rate of response, we gathered data from practitioners (workers/employees) who utilize digital integration and can effectively complete the questionnaires related to digital integration.

We administered the questionnaires via the Tillion Survey (Tillion is the survey platform with the biggest panel in South Korea. With consistent and strict panel management, they can manage and administer professional and sophisticated surveys to meet various client needs) in the fall of 2022. In total, 4564 questionnaires were disbursed, and 236 responses (5.17%) were received. Following the screening for incomplete and unreliable data, the final sample size consisted of 222 responses (4.85%).

Referring to the Korean standard industrial classification, the industry distribution of the respondents is in the order of 'machinery and equipment' (17.6%), 'automotive-related' (15.3%), 'furniture' (13.5%), and 'electronics' (11.3%). The sizes of the respondents' companies by the number of employees are 'more than 300' (29.3%), '100~299' (18.5%), and '50~99' (13.1%). Therefore, it seems that companies that are large enough to adopt digital technologies have mainly responded. The informants' profiles show that they are 'mid-level managers (including department manager, team manager, and assistant manager)' (68%), 'general employees' (28.4%), and 'executives or CEOs' (3.7%). Regarding the employment period, 35.6% of informants responded that they had worked for their company for 'more than 10 years', and 30.6% responded 'from 5 to 10 years'.

### 3.2. Measures

The survey instrument was developed based on the pertinent supply chain management literature. The questionnaire was originally composed in English and carefully translated into Korean to ensure consistency of meaning between both versions and to mitigate comprehension issues. The translated questionnaire was confirmed through interviews with corporate executives. We delineated measurements for each construct, and the survey was implemented through Qualtrics. In the survey, we included a question that filtered out participants who were not employees of manufacturing companies.

Table 1 shows all survey items of this study. All of the measurement items are assessed using a 7-point Likert scale, with response options ranging from '1' (Strongly disagree) to '7' (Strongly agree). Generalized measures of digital transformation between firms are not yet established in the existing literature. We conducted a thorough review of literature related to digital transformation [75–77] and modified the eight survey items to suit the circumstances of domestic companies. We checked the modified survey items to see whether the meanings of the concepts were accurately conveyed and whether they

were easy for corporate practitioners to understand. To measure information sharing, we drew upon three items [78–80]. Time-based performance is a second-order structure with three terms: time-to-market, measured by the extent to which new product development time and new product introduction time have been accelerated; time-to-product, measured by the extent to which procurement lead time and manufacturing lead time have been shortened; and responsiveness, measured by the extent to which production support time and responsiveness to customers have been accelerated. The measures are drawn from reference [58–60]. Respondents were also asked to provide subjective ratings of their firm's performance in comparison to their performance before the implementation of digital systems. For developmental (flexible and external) corporate culture, we derived four items from reference [64]. B2B trust is measured with three items from reference [74].

**Table 1.** Measurement items and reliability.

| Variables | Measurement Items | Factor Loading | Cronbach's α | CR [1] |
|---|---|---|---|---|
| Digital transformation | **The IT system introduced in our company helps with the following:**<br>1. market information sharing, 2. demand forecast sharing,<br>3. product/service feedback sharing from distributors<br>4. production planning sharing with distributors<br>5. supply plan (procurement plan) from supplier<br>6. production planning sharing to supplier<br>7. participation in production and procurement processes,<br>8. participation in the product development process of supplier | 0.854~ 0.924 | 0.968 | 0.925 |
| Information sharing | **Our company and partners:**<br>1. provide each other with information about events or changes that may affect each other.<br>2. provide important information to each other.<br>3. share information and knowledge about key business procedures. | 0.892~ 0.903 | 0.925 | 0.871 |
| Time-based performance | **Time-to-market:**<br>our company's 1. product development time and 2. product launch time have become faster.<br>**Time-to-product:**<br>our company's 1. procurement time (supplier lead time, transportation, warehousing, inspection time, etc.) and 2. production time (from order request to production) have become faster<br>**Responsiveness**:<br>1. our company's 1. product support time and 2. customer response time have become faster. | 0.883~ 0.936 | 0.955 | 0.952 |
| Corporate culture | **Our company:**<br>1. pursues a dynamic and challenging entrepreneurship<br>2. takes risks.<br>3. emphasizes innovation throughout our business operations.<br>4. seeks to take the lead in launching new products/services. | 0.805~ 0.926 | 0.929 | 0.840 |
| B2B trust | **Our company:**<br>1. trusts our supply chain partners.<br>2. believes that our partners will fulfill their responsibilities.<br>3. believes that our partners are sincere in their efforts to maintain relationships with us. | 0.891~ 0.948 | 0.939 | 0.895 |

[1] Composite reliability.

### 3.3. Measurement Model

As shown in Figure 1 below, this study seeks to verify the three-step causal relationship and comprehensively identify the mediating and moderating effects. For our analysis, we employ structural equation modeling (using AMOS 17.0), a method capable of delineating causal relationships among all paths within the overarching model, thus illustrating the relative impacts of these causal connections. Prior to analysis, we assess the adequacy of the measurement model using the AMOS and PASW statistics programs. The results regarding the reliability and validity of the variables are presented in Tables 1 and 2, respectively. Regarding the reliability of the variables, Cronbach's alpha values exceed the recommended threshold of 0.70 [81], indicating satisfactory internal consistency. Additionally, standardized factor loadings for all items are higher than the 0.70 benchmark [82], further affirming the reliability of the measures. Next, to ensure the validity of our measures for structural equation modeling, we assess both convergent and discriminant validity. Convergent validity is evaluated using two criteria: (1) an average variance extracted (AVE) exceeding 0.50 and (2) a composite reliability (CR) of 0.70 or higher for each construct [81]. Discriminant validity is confirmed by verifying that the AVE of each latent construct surpassed the square of its correlation with other variables [83].

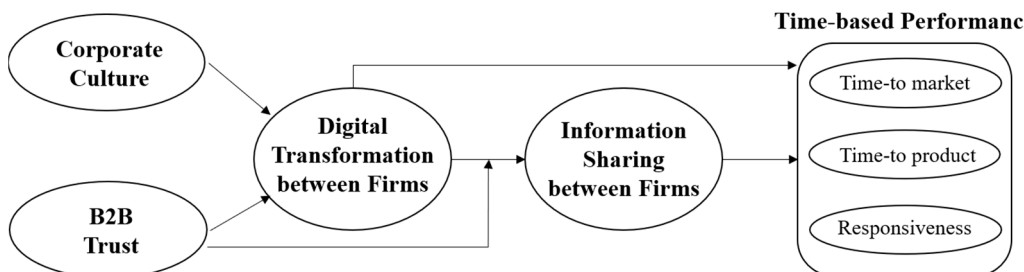

**Figure 1.** Research model.

**Table 2.** Convergent/discriminant validity and correlations.

| Variables | Mean | S.D. | AVE [1] | $\sqrt{AVE}$ | 1 | 2 | 3 | 4 | 5 |
|---|---|---|---|---|---|---|---|---|---|
| 1 | 4.17 | 1.43 | 0.606 | 0.779 | 1 | | | | |
| 2 | 4.20 | 1.27 | 0.693 | 0.832 | 0.715 ** | 1 | | | |
| 3 | 4.28 | 1.29 | 0.868 | 0.932 | 0.709 ** | 0.798 ** | 1 | | |
| 4 | 3.99 | 1.45 | 0.567 | 0.753 | 0.699 ** | 0.710 ** | 0.705 ** | 1 | |
| 5 | 4.59 | 1.28 | 0.740 | 0.860 | 0.640 ** | 0.684 ** | 0.720 ** | 0.730 ** | 1 |

1. Digital transformation, 2. information sharing, 3. time-based performance, 4. corporate culture, and 5. B2B trust. ** $p < 0.01$. [1] Average variance extracted.

Using confirmatory factor analysis (CFA), the overall fit index of the measurement model is $\chi^2 = 437.431$, *df* = 239, $\chi^2/df$ = 1.83, *p* = 0.00, SRMR = 0.03, IFI = 0.969, CFI = 0.968, and RMSEA = 0.06, which is an acceptable level. In detail, the overall fit indexes of the measurement model should be as follows [84–86]: (1) $\chi^2/df$ should be smaller than 3~5; (2) the standardized root mean square residual (SRMR) should be under 0.08; (3) the comparative fit index (CFI) and incremental fit index (IFI) should be 0.90 or greater; and (4) the root mean square error of approximation (RMSEA) should be smaller than 0.10.

## 4. Results

### 4.1. The Impact of Digital Transformation on Time-Based Performance

The fit index of the research model indicates a good fit as $\chi^2 = 528.481$, *df* = 243, $\chi^2/df$ = 2.175, *p* = 0.00, IFI = 0.955, CFI = 0.954, and RMSEA = 0.06, which is an acceptable level.

For H1, which assumes the mediating effect of information sharing, standardized β are 0.76 (digital transformation → information sharing, *p* = 0.00) and 0.67 (information sharing → performance, *p* = 0.00) (see Table 3). In order to verify the mediation effect assumed in

the hypothesis, additional analysis is conducted, and the results are shown in Table 4. As a result, all path coefficients from Model 1 to Model 3 are significant, and the difference between Model 2, which restricts the direct effect of digital transformation to zero, and Model 3, which includes it, is significant ($\Delta\chi^2(df)$ = 6.63(1), $p$ < 0.01). Furthermore, the direct effect in Model 1 (unstandardized β = 0.70, $p$ = 0.00) is significantly reduced in Model 3 (unstandardized β = 0.21, $p$ = 0.00), which includes the mediation effect, confirming the existence of a partial mediating effect of information sharing. The results support H1. In other words, introducing a digital communication system for B2B transaction management improves corporate performance by activating information sharing between the two companies.

**Table 3.** Results of hypotheses.

| | Path | | Standardized β (Unstandardized β) | S.E. | C.R. | Result |
|---|---|---|---|---|---|---|
| Corporate culture | → | Digital transformation | 0.550 *** (0.534) | 0.084 | 6.325 | H3. Support |
| B2B trust | → | Digital transformation | 0.252 ** (0.286) | 0.094 | 3.057 | H4. Support |
| Digital transformation | → | Information sharing | 0.762 *** (0.698) | 0.057 | 12.300 | H1. Support |
| Information sharing | → | Time-based performance | 0.669 *** (0.695) | 0.076 | 9.108 | |
| Digital transformation | → | Time-based performance | 0.239 *** (0.227) | 0.065 | 3.517 | H2. Support |

*** $p$ = 0.00, ** $p$ < 0.01.

**Table 4.** Mediation effect analysis.

| | Path | | Model 1 [1] | Model 2 [2] | Model 3 [3] |
|---|---|---|---|---|---|
| | | | | Standardized β (Unstandardized β) | |
| Digital transformation | → | Time-based performance | 0.733 *** (0.701) | | 0.219 *** (0.210) |
| Digital transformation | → | Information sharing | | 0.769 *** (0.708) | 0.750 *** (0.691) |
| Information sharing | → | Time-based performance | | 0.864 *** (0.900) | 0.686 ** (0.714) |

** $p$ < 0.01, *** $p$ = 0.00; [1] $\chi^2$ = 168.912—$df$ = 73, [2] $\chi^2$ = 250.806—$df$ = 114, [3] $\chi^2$ = 240.373—$df$ = 113.

The result of the analysis for H2 that digital transformation has a direct effect on performance was a standardized β of 0.24 ($p$ = 0.00), supporting the hypothesis (see Table 3). This confirms the assumption that the digitized information created during the digital transformation process is distributed efficiently within the company, which has an additional direct effect on corporate performance.

### 4.2. Antecedent Effects of Corporate Culture and B2B Trust on Digital Transformation

H3, which states that a more developed corporate culture is more likely to accept digital transformation, is supported by a significant standardized path coefficient of 0.55 ($p$ = 0.00) (see Table 3). This finding suggests that companies with a flexible and challenging corporate culture are receptive to digital transformation.

Regarding the positive antecedent effect of B2B trust on digital transformation, a statistically significant path coefficient was confirmed (standardized β = 0.25, $p$ = 0.00), supporting H4 (see Table 3). In other words, the more companies trust each other, the more time and effort they spend on innovation, which is called digital transformation.

### 4.3. Moderating Effects of B2B Trust on Relationship between Digital Transformation and Information Sharing

The fit index of the research model indicates a good fit as $\chi^2$ = 868.861, $df$ = 486, $\chi^2/df$ = 1.788, $p$ = 0.00, IFI = 0.923, CFI = 0.922, and RMSEA = 0.06, which is an acceptable level. Table 5 shows the results of the moderating effect with multigroup analysis. We expect that B2B trust positively affects the relationship between digital transformation and information sharing. We divided respondents into two groups—low trust and high

trust—based on the average B2B trust and compared the differences between groups on the corresponding path (digital transformation → information sharing).

**Table 5.** Moderating effect analysis.

| Path | | | Standardized β (Unstandardized β) | S.E. | C.R. | Result |
|---|---|---|---|---|---|---|
| Digital transformation | → | Information sharing | Low trust 0.687 *** (0.806) | 0.116 | 6.960 | H5. Reject |
| | | | High trust 0.647 *** (0.418) | 0.067 | 6.258 | |

Unconstrained model, $\chi^2$ = 868.861, *df* = 486; structural weights model, $\chi^2$ = 878.034, *df* = 487; *** *p* = 0.00.

The results indicate that the regression weight of each group is β = 0.81 (low trust group, *p* = 0.00) and β = 0.42 (high trust group, *p* = 0.00), and the difference between these two groups is significant ($\Delta\chi^2$(*df*) = 9.17(1), *p* < 0.01). This means that more information exchange occurs through digital transformation systems in corporate relationships with low (vs. high) trust, which is the exact opposite of the prediction of H5. Thus, the results reject H5.

There are two possible interpretations of this result. First, in corporate relationships with high trust, there is a high possibility that there will already be familiar cooperation and communication methods (e.g., friendly phone calls between managers, regular face-to-face meetings, etc.). In other words, because information sharing is sufficiently accomplished through existing methods, the effect of information sharing through the digital conversion system may not be as expected. On the other hand, companies with low trust may exchange information effectively through these digital systems because the people in charge are not familiar with each other, or the transaction period is short, so communication is not familiar.

## 5. Discussion

This study explores under what conditions a company's digital transformation strategy is well accepted and how it affects company performance. We verify this research model through data from 222 Korean manufacturing companies and interpret the results as follows.

The findings of this study elucidate the direct impact of digital transformation on corporate performance (H2) while also confirming the mediating role of information sharing until digital transformation leads to market leadership, achieves efficient production management, and enhances customer satisfaction (H1). These findings underscore the importance of integrating digital technology into operational processes as a pivotal corporate strategy [16]. Moreover, they reaffirm prior research indicating that digital systems that facilitate accurate and comprehensive information sharing contribute to enhancing a company's competitiveness [11,47]. While previous studies have examined these relationships individually, our study stands out by conceptualizing the process of how digital transformation enhances corporate performance through the mediating effect of information sharing within a single model.

Next, we discovered that innovative technologies, such as digital transformation, are embraced within a flexible and dynamic organizational culture (H3). This finding implies the need to investigate the conditions necessary for companies to apply and leverage innovation strategies internally. Previous studies on organizational culture have categorized and defined it based on the organization's attitude towards future strategies or its response to uncertainties in the business environment [27,62,64]. This suggests an inherent association between unpredictable changes, namely innovation, and organizational culture. From this perspective, our study confirms the findings predicted by prior research, indicating that the acceptance of digital transformation is smoother within a developmental organizational culture.

Another antecedent variable considered in the study, B2B trust, also exhibits a positive antecedent effect concerning digital transformation (H4). We find that when there is trust

among supply chain members, they make greater efforts and investments to introduce innovative systems. According to the agency theory, firms inherently prioritize their own interests [28,71,72]. Therefore, inter-firm trust has been extensively explored as a crucial variable in inter-firm transaction performance, with nearly all studies indicating a positive impact [73]. Consequently, the research finding that new investments in applying digital transformation to inter-firm transactions are more seamlessly and actively carried out within high-trust business relationships is highly valid.

Lastly, we assumed (H5) that as B2B trust increases, information sharing through digital transformation would be enhanced. However, our research findings reveal the opposite. The results indicate that the effectiveness of digital communication methods may actually increase when supply chain members lack familiarity or trust has not been established. This finding warrants attention. A trustworthy company typically possesses a well-established method for exchanging information, even in the absence of digital technology. Consequently, the improvement effect of digital technology on information quality may not be readily apparent. Put differently, the greater efficiency and effectiveness of digital information exchange in corporate relationships characterized by shorter transaction periods or limited familiarity suggest the need to actively consider the introduction of digital transformation technology, particularly when engaging in important transactions with new companies.

## 6. Conclusions

### 6.1. Implications

This study has theoretical implications in that it presents various antecedent and mediating variables for the important corporate strategy of digital transformation and assumes and verifies their roles and effects based on existing research. In particular, we verified the mediating effect that corporate performance increases by promoting information sharing through the introduction of digital transformation among supply chain members. This is significant because existing studies on digital transformation point to the importance of information sharing when explaining its effects, and this influence relationship has been demonstrated with corporate data. In addition, new survey items were developed to measure the digital transformation system used in supply chain transaction situations, which also has academic significance. In relation to efficiency, which is the focus of existing digital transformation research, there is meaning in presenting and verifying a research model using the concept of time-based performance as a sustainable performance indicator for manufacturing companies.

This study has academic significance in that it demonstrated a research model with data from 222 Korean manufacturing companies. In particular, the Republic of Korea has a very well-equipped digital infrastructure at a world-class level, so people, as well as companies, have a high level of familiarity and understanding of IT technology. For example, out of the Republic of Korea's total retail market size of USD 317 billion in 2022, the amount of online (internet and mobile) shopping amounted to USD 158 billion (the Republic of Korea Chamber of Commerce and Industry '2023 Distribution and Logistics Statistics Collection'). In addition, due to the narrow geographical conditions, an efficient logistics system such as same-day delivery greatly determines the survival and competitiveness of distribution companies, making it a business environment in which digital innovation is essential across the entire supply chain, including manufacturing companies and supply companies. These facts mean that the manufacturing companies surveyed in this study have been exposed to a lot of digital transformation strategies or are already using them and that respondents were also able to answer accurately enough about the role and level of use of this system.

From a practical perspective, the findings of this study suggest that companies that want to introduce digital transformation or those that are having difficulties in the process of introducing it should check their corporate culture and relationships with partner companies. Digital transformation is an innovation strategy that requires significant cost,

time, and effort. Therefore, a company must be well prepared to accept this strategy, and to do so, the company must check whether the organizational culture is flexible and ready to accept external changes. This perspective applies not only to one's own company but also to partner companies that use this digital system. Considering that supply chain efficiency will increase the sustainability of all companies, including manufacturing companies, in the future, the use of digital transformation strategies for the supply chain is inevitable. In other words, when forming a new supply chain or changing partner companies, it will be necessary to review the company's organizational culture. In particular, if trust between partner companies is not established, it may be difficult to introduce digital transformation in supply chain management. However, this problem can be alleviated if the partner has a very flexible organizational culture.

Finally, although we confirm a conclusion that is different from the hypothesis of this study, the result has great practical implications. Specifically, trust between companies is a preceding variable that leads to the acceptance of digital transformation but does not promote the information-sharing effect of digital transformation. This finding provides guidance to corporate executives that they should actively consider introducing a digital transformation strategy if they are conducting an important transaction with a new business they are not familiar with. Digital-based communication eliminates the awkwardness of face-to-face contact or formal greetings, enabling smooth and sufficient information exchange from the beginning of a transaction. In other words, unlike a high-trust relationship where there is already a familiar communication method, a digital-based communication system can be a much more effective means of information sharing in low-trust corporate transactions.

### 6.2. Limitations and Suggestions

This study has several limitations that suggest directions for future research. First, we conducted a survey targeting only manufacturing companies among supply chain members. This is because manufacturing companies that produce products and services are at the center of the supply chain. However, as the influence of distribution companies on the market has grown significantly in recent years, research from the perspective of distribution companies will also be necessary in the future. In addition, this study collected data from manufacturing companies in various industries, so it will be necessary to compare the research results depending on whether companies produce final products under their own brand, subcontractors, or OEM companies in future research.

Second, this study developed a research model on the effects of digital transformation using trust, organizational culture, and information sharing as variables, but these are not all of the important strategic variables discussed in many previous studies on SCM. Therefore, there is a need to explore other B2B relationship variables or company characteristic variables in future research. For example, in addition to trust, transaction period or transaction importance can be studied as moderating variables.

Finally, we explore the effects of digital transformation among companies. However, digital transformation can also be applied within a company and used as a means of communication between departments and workers. Future researchers can investigate whether this research model can be applied inside a company and what similarities and differences exist between digital transformation within and between companies.

**Author Contributions:** J.W.K. suggested the initial research idea and developed the research model. C.H.P. performed the literature review and designed and executed the survey. J.H.R. conducted the literature review, analyzed the data, and will be the primary party handling the review process. All authors have read and agreed to the published version of the manuscript.

**Funding:** This research is partially supported by a Korea University Business School Research Grant.

**Institutional Review Board Statement:** In accordance with the Republic of Korea's research ethics laws, the survey in this study is not subject to legal permissions or regulations. In a written statement on the first page of the questionnaire, respondents were informed about the objectives of the study,

that the study was confidential, and that it was being conducted for scientific purposes only. Additionally, the survey was computer-based and allowed respondents to stop or refuse to answer at any time. Therefore, only those who were willing to participate completed the questionnaire.

**Informed Consent Statement:** Informed consent was obtained from all survey participants.

**Data Availability Statement:** The datasets used and/or analyzed during the current study are available from the corresponding author upon reasonable request.

**Acknowledgments:** This paper has been revised and updated by extracting the author's (Chul Hung Park's) master's thesis.

**Conflicts of Interest:** Author Chul Hung Park was employed by Mahlkonig Korea. The remaining authors declare that the research was conducted in the absence of any commercial or financial relationships that could be construed as a potential conflict of interest.

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
