# Peer review of "How Does Digital Transformation Improve Supply Chain Performance: A Manufacturer’s Perspective"

_sustainability, doi:10.3390/su16073046_

Round 1

Reviewer 1 Report

Comments and Suggestions for Authors

This paper focuses on the integration of digital technologies within corporate strategic frameworks for digital transformation. The study aims to explore how digital transformation technologies are utilized among supply chain members. The research collected and analyzed surveys from 222 Korean manufacturing companies to verify the research hypotheses. Five hypotheses were examined, addressing both positive and negative aspects of the subject. However, at the end of the introduction, the authors redundantly present the same results, which may not be necessary in this section.

The paper exhibits several issues:

i) It relies heavily on outdated references, with only 25% being more recent (approximately 74% are before 2019).

ii) It lacks specification of the survey date. The timing of data collection is unclear.

iii) The authors employ a qualitative scale as a quantitative method. It might be beneficial to explain the use of qualitative data and its application in quantitative statistical methods.

iv) Table 3 is not explained in the text.

v) The text and tables contain mistakes (e.g., see Table 4 and 5.1). The formatting of the list in "Summary of Findings and Implications" is incorrect.

vi) The Sample and Data Collection method is not clearly described. Was random sampling used? Did the sample adequately represent the population?

vii) The rotation method applied is not specified.

viii) The discussion section is overly generic. It would be more insightful to explain the results for each hypothesis individually.

ix) There is no statistical formulation of hypotheses.

x) Including a section on general conclusions would be beneficial. The authors could address limitations of the work therein.

The issue is not novel, but the paper offers both practical and theoretical contributions.

Comments on the Quality of English Language

There are the some mistakes in the tables and text. I recommend including a section of conclusions."

Reviewer 2 Report

Comments and Suggestions for Authors

Dear Authors,

I read your paper with interest and felt the content timely on digital transformation. The fact that one of the authors did preliminary work on a master’s thesis is a high merit. You have reviewed a good range of academic literature pertinent to the themes and variables of your research. The empirical work is equally sound for regression analysis, albeit a relatively small sample size. The construction of the structural model (Research Model) could have been improved by re-arranging the factors/constructs. The Korean version of the questionnaire has been tested by interviews with corporate executives (ln 267 p. 6). Further elaboration, need not be long, is desirable, i.e. the number of interviews conducted and any changes have been made as a result? Using AMOS is efficient for SEM analysis, however inserting a measurement model could help illustrate the regression paths. In sum the paper as it stands now contain a number of major, especially conceptual clarity, as well as minor concerns, typos and spellings, which you need to deal with.

Firstly, the concept of corporate sustainability (ln 41 p. 1) needs to be defined and discussed. What do you mean by corporate sustainability and how this is related to your research on digital transformation?

Secondly, when you say you categorise corporate culture into developmental and hierarchical cultures, does this mean you coin the terms or concepts (ln 197 p. 5)? Even so you need to provide some academic literature as prerequisite of a new definition.

Thirdly, in the measurement of B2B trust you divided it into two sub-groups, Low and High (ln 366 p. 10). Again is this your typology or adopted from the literature. Either way you need to provide explanations and better still definitions.

Fourthly, your sampling population. You acknowledged the deficiency of selecting only manufacturing companies. Even if you chose only the manufacturing companies, the analysis could be enriched by differentiating the sample into OEMs and components suppliers in tier one, two, etc.

On the minor concerns, there are a few as listed below:

1.       The use of apostrophe need attention (ln 33 p. 1)

2.       Remove the word ‘chapter’ (ln 90 p. 2), as this is no longer part of your thesis, rather using section in this paper

3.       Use plural of hypotheses rather than hypothesis (ln 77 p. 2), otherwise the reader would think you had only one hypothesis to be tested

4.       For hypothesis formulation just comply write H1 instead of the cumbersome version of Hypothesis 1 (H1)

5.       A major mistake in Table 1 on Responsiveness

I hope these comments help you improve the paper for publication.

Comments on the Quality of English Language

The paper is well-strucutred, and relatively well-written with some minor corrections.

Reviewer 3 Report

Comments and Suggestions for Authors

Comments and Suggestions for Authors

The study aims to explore how digital transformation technologies used between supply chain members can be well introduced and utilized. The subject of the paper is in line with the aims and scope of the Journal, but this connection is not well-elaborated. The authors should correct the paper regarding the following comments.

1.     The connection of the paper with the aims and scope of the Journal is weak. The authors did not make clear connection of the studied problem with any of the three main sustainability pillars (environmental, economic and social).

2.     The abstract is not written well. The authors did not provide a background of the study enough to understand their motivation for this problem. They also didn’t highlight the methodology they used, or the main contributions of the paper.

3.     Theoretical background (literature review) is comprehensive, bust mostly outdated. The authors used only several sources from the last three years. The authors should also try to highlight the research gaps in the section 2.

4.     The authors did not provide a literature review regarding all aspects of their study. They focused solely on the variables and hypothesis.

5.     My biggest concern regarding this paper is its scientific contribution. The authors did invest a substantial effort to collect the data, but they simply applied well-established methodology for processing these data. Also, I’m afraid that the results and conclusions are case sensitive. It is highly questionable if the conclusions are universal.

6.     Authors did not discuss how the results can be interpreted in perspective of previous studies. Discussion should clearly and concisely explain the significance of the obtained results in order to demonstrate the actual contribution of the article to this field of research, when compared with the existing and studied literature.

7.     English writing should be improved. There are grammar, syntax and style errors that need to be addressed.

8.     Some technical issues should be addressed:

a)     There should be at least a couple of sentences between the heading of different levels (e.g. between the headings 2 and 2.1).

b)     The authors did not cite the references in the text according to the Instructions for authors.

c)     All tables must be mentioned somewhere in the main text (e.g. Table 3 is not mentioned anywhere).

d)     References in the reference list are not formatted according to the Instructions for authors.

e)     Abbreviations and acronyms must be defined the first time they appear in the text. For example, abbreviation “B2B” is not defined. Check the rest of them.

Comments on the Quality of English Language

1.     English writing should be improved. There are grammar, syntax and style errors that need to be addressed.

Reviewer 4 Report

Comments and Suggestions for Authors

Thank you for providing me with the opportunity to review your article. Congratulations on the commendable effort you've put into your work thus far.

Upon careful examination, I've observed several compelling arguments within your article that lack proper references. It is crucial to address this discrepancy, as substantiating claims with credible sources strengthens the overall validity of your work.

On the flip side, a notable weakness of your article lies in the age of its references. I noticed that the majority of your citations stem from the 1980s and 1990s, with only a solitary reference from 2022 and two from 2023. Despite the technological focus of your article, relying predominantly on dated sources creates a discrepancy between the subject matter and the currency of your references.

From my perspective, this presents a significant concern. Updating and diversifying your references are imperative to ensure that your article remains pertinent and rooted in contemporary research. The rapid evolution of technology necessitates a continuous renewal of references to mitigate the risk of relying on outdated or "deprecated" arguments.

In summary, addressing these issues will enhance the credibility and relevance of your article, aligning it more closely with current advancements in the field.

A few examples:

Lines 31-23: Strong argument without a reference.

Lines 35-37: Also a strong argument without reference.

Line 40: I am not sure if this is the format MDPI expects for the reference.

Lines 40-43: This sentence deserves a reference.

Line 45: COVID contributed to speed up the process, but unsure we can state workers became accustomed because of COVID only.

Lines 48-52: It deserves a reference.

Round 2

Reviewer 3 Report

Comments and Suggestions for Authors

The authors have successfully addressed all issues from the previous review round, thus significantly improving the quality of their paper. Therefore, I suggest an acceptance of the paper in its present form.

Reviewer 4 Report

Comments and Suggestions for Authors

Thank you for meeting my suggestions.